# Effects of Loganin on Bone Formation and Resorption In Vitro and In Vivo

**DOI:** 10.3390/ijms232214128

**Published:** 2022-11-16

**Authors:** Chang-Gun Lee, Do-Wan Kim, Jeonghyun Kim, Laxmi Prasad Uprety, Kang-Il Oh, Shivani Singh, Jisu Yoo, Hyun-Seok Jin, Tae Hyun Choi, Eunkuk Park, Seon-Yong Jeong

**Affiliations:** 1Department of Medical Genetics, Ajou University School of Medicine, Suwon 16499, Republic of Korea; 2Department of Biomedical Sciences, Graduate School of Medicine, Ajou University, Suwon 16499, Republic of Korea; 3AI-Superconvergence KIURI Translational Research Center, Ajou University School of Medicine, Suwon 16499, Republic of Korea; 4Department of Medical Sciences, Graduate School of Medicine, Ajou University, Suwon 16499, Republic of Korea; 5Department of Biomedical Laboratory Science, College of Life and Health Sciences, Hoseo University, Asan 31499, Republic of Korea; 6Department of Molecular Imaging, Korea Institute of Radiological and Medical Sciences, Seoul 01812, Republic of Korea

**Keywords:** loganin, bioactive compound, anti-osteoporosis agent, bone remodeling

## Abstract

Osteoporosis is a disease caused by impaired bone remodeling that is especially prevalent in elderly and postmenopausal women. Although numerous chemical agents have been developed to prevent osteoporosis, arguments remain regarding their side effects. Here, we demonstrated the effects of loganin, a single bioactive compound isolated from *Cornus officinalis*, on osteoblast and osteoclast differentiation in vitro and on ovariectomy (OVX)-induced osteoporosis in mice in vivo. Loganin treatment increased the differentiation of mouse preosteoblast cells into osteoblasts and suppressed osteoclast differentiation in primary monocytes by regulating the mRNA expression levels of differentiation markers. Similar results were obtained in an osteoblast–osteoclast co-culture system, which showed that loganin enhanced alkaline phosphatase (ALP) activity and reduced TRAP activity. In in vivo experiments, the oral administration of loganin prevented the OVX-induced loss of bone mineral density (BMD) and microstructure in mice and improved bone parameters. In addition, loganin significantly increased the serum OPG/RANKL ratio and promoted osteogenic activity during bone remodeling. Our findings suggest that loganin could be used as an alternative treatment to protect against osteoporosis.

## 1. Introduction

Osteoporosis is a skeletal disorder caused by a reduction in bone mineral density (BMD), leading to a high risk of bone fractures [1]. The major cause of osteoporosis is the dysfunction of bone remodeling, which is regulated by the balance between bone matrix formation by osteoblasts and the resorption of old bone by osteoclasts [2]. These two major bone cells communicate with each other via osteoprotegerin (OPG)/receptor activator of nuclear factor kappa B ligand (RANKL) signaling to sustain skeletal integrity [3]. It has been reported that hormones such as estrogen and parathyroid hormone play a major role in bone remodeling by regulating OPG/RANKL expression in osteocytes [4,5].

The prevalence of osteoporosis is high in older women due to an estrogen imbalance caused by menopause [6]. Considering that approximately 50% of postmenopausal women suffer from osteoporosis, the management of postmenopausal osteoporosis is needed to improve the quality of life in aging populations globally [7]. Many pharmacological therapies have been developed to prevent osteoporosis by improving bone remodeling [8]. Several medications, such as selective estrogen receptor modulators (raloxifene) and anti-resorptive agents (bisphosphonate) have been used to treat osteoporosis; however, the long-term use of these chemical agents is associated with adverse effects, including swallowing, irritation, increased risk of blood clots, stomach pain, atypical femoral fractures and osteonecrosis of the jaw [9,10]. Because of these complications, another mechanism targeting bone remodeling such as sclerostin and parathyroid hormone have been considered as alternative therapies to prevent osteoporosis [11,12].

Phytochemical compounds derived from herbal plants have been used as alternative therapies because of their efficacy and association with few adverse effects [13]. In recent years, herbal plants have been reported to contain a variety of bioactive constituents that have diverse effects on numerous diseases [14]. *Cornus officinalis*, also known as shanzhuyu, is prevalent and traditionally used in East Asia because of its protective effects against skeletal, cardiovascular, and liver diseases and immunomodulation [15]. One study demonstrated that *Cornus officinalis* contains numerous bioactive compounds, including loganin, morroniside, ursolic acid, oleanolic acid, and cornuside [16]. In addition, a previous study revealed that *Cornus officinalis* promotes anti-postmenopausal effects in ovariectomized (OVX) mice, indicating that potential bioactive compounds in *Cornus officinalis* enhance the protective effects against OVX-induced postmenopausal osteoporosis [17].

Loganin, an iridoid glycoside, is a major component of *Cornus officinalis* and has potent effects on neuroinflammation, colitis, glomerulonephritis, and cancer [18]. A previous study revealed that loganin promotes osteoblastic differentiation by inhibiting apoptotic cell death in preosteoblastic MC3T3-E1 cells, suggesting that loganin is a potential anti-osteoporotic agent [19]. However, detailed studies on the inhibitory effects of loganin on osteoclast activity and osteoporotic mouse models have not yet been conducted.

In the present study, we examined the effects of a single bioactive compound, loganin, on osteoblast and osteoclast differentiation in vitro and in mice with osteoporosis in vivo. Osteoblast and osteoclast differentiation were evaluated using the mouse preosteoblast cell line MC3T3-E1 and primary cultured bone marrow monocytes, respectively. Finally, an in vivo assessment of the anti-osteoporosis effect of loganin was performed in an OVX-induced osteoporosis mouse model.

## 2. Results and Discussion

### 2.1. Loganin Increased the Differentiation of Mouse Preosteoblasts into Osteoblasts

Osteoblasts are responsible for the formation of new bone, which is mediated by the synthesis of bone matrix [20]. During this process, alkaline phosphatase (ALP) is highly expressed in the membranes of osteoblasts and initiates osteoblastic mineralization [21]. Therefore, ALP expression in osteoblasts is considered a biomarker of bone formation for evaluating osteoblast differentiation [22]. To investigate whether loganin promotes osteoblast differentiation, MC3T3-E1 cells were induced with osteoblast differentiation medium in the presence of different concentrations of loganin (1, 5, and 10 μM) for 3 days. Loganin did not affect cellular viability during the 3 days of treatment (Figure 1A). However, increased ALP activity was observed in cells treated with 5 and 10 μM loganin compared with the induction group (Figure 1B). Consistently, the number of ALP-positive cells was increased by loganin administration in a dose-dependent manner (Figure 1C). In addition, we determined the promotive effect of loganin on osteoblast differentiation by evaluating osteogenic markers such as *Alpl*, bone gamma-carboxyglutamate protein (*Bglap*)*,* and runt-related transcription factor 2 (*Runx2*). *Alpl* encodes ALP and enhances osteoblast differentiation [23]. *Bglap*, also known as osteocalcin, is a calcium-binding protein secreted from osteoblasts that concentrates calcium in the bone [24]. *Runx2* is a transcription factor that plays an essential role in initiating osteoblast differentiation [25]. These genes are potential biomarkers of osteoblast differentiation [26]. Quantitative reverse-transcriptase polymerase chain reaction (qRT-PCR) analysis showed that loganin treatment elevated the mRNA expression levels of osteoblast-inducing markers, including *Alpl*, *Bglap*, and *Runx2* (Figure 2). These results indicate that loganin promotes osteoblast differentiation by increasing the mRNA expression of osteogenesis-related markers in mouse preosteoblasts.

### 2.2. Loganin Inhibited Differentiation of Mouse Primary Monocytes into Osteoclasts

Increased osteoclast differentiation and activity is a critical event leading to excessive bone loss and vulnerability to fragility, resulting in the development of osteoporosis [27]. Previous studies have demonstrated that anti-osteoporotic agents prevent osteoclast differentiation and activity [28]. Tartrate-resistant acid phosphatase (TRAP) is a glycosylated monomeric metalloprotein enzyme expressed in osteoclasts that promotes the dephosphorylation of bone matrix proteins involved in bone formation, such as osteopontin and sialoprotein [29]. Therefore, many studies have suggested that the inhibition of TRAP activity during osteoclast differentiation is an important factor in the prevention of osteoporotic bone loss [30,31,32]. To investigate the effects of loganin on osteoclast differentiation in vitro, primary cultured bone marrow monocytes were differentiated into osteoclasts in the presence of osteoclast-stimulating factors, including macrophage/monocyte stimulating factor (M-CSF) and RANKL for 5 days. Successfully isolated monocytes derived from bone marrow were confirmed by fluorescence-activated cell sorting (FACS) analysis using a monocyte specific cluster of differentiation 11b (CD11b) antibody (Figure 3A). Loganin did not affect the viability of monocytes after 5 days of incubation (Figure 3B). However, loganin treatment significantly decreased the TRAP activity in a dose-dependent manner (Figure 3C). In addition, the number of TRAP-positive multinucleated cells were decreased by loganin treatment (Figure 3D). Then, the mRNA expression levels of osteoclast-associated genes, such as acid phosphatase 5 (*Acp5*), cathepsin K (*Ctsk*), and matrix metalloproteinase 9 (*Mmp9*) were examined by qRT-PCR. *Acp5* is expressed in osteoclasts and encodes TRAP [33]. Cathepsin K and *Mmp9* are proteases secreted by osteoclasts that are involved in collagen degradation during bone resorption [34]. After 5 days of osteoclast differentiation, loganin decreased the mRNA expression levels of *Acp5*, *Ctsk*, and *Mmp9* (Figure 4). These results suggest that loganin inhibits osteoclast differentiation and bone resorption by decreasing the mRNA expression levels of *Acp5*, *Ctsk*, and *Mmp9*.

### 2.3. Loganin Decreased Osteoclast Activity in an Osteoblast-Osteoclast Co-Culture System

During bone remodeling, communication between the osteoblasts and osteoclasts plays a critical role in the regulation of cellular activities and differentiation through direct cell–cell contact, secretory mediators, and interactions with the extracellular matrix [35]. In this regard, an osteoblast-osteoclast co-culture system provides a similar environment in vitro and is a useful methodology for evaluating the process of bone remodeling [36]. To examine the effect of loganin on the osteoblast–osteoclast co-culture system, mouse primary monocytes were co-cultured with MC3T3-E1 cells for 24, 48, 72, and 120 h in the presence of loganin (10 μM). Osteoblast and osteoclast differentiation were assessed based on ALP/TRAP activity. Higher ALP activity was observed in the co-culture system at 72 h compared to the MC3T3-E1 monoculture (Figure 5A). However, the ALP activity in the loganin treatment group peaked at 48 h; this was earlier than peak ALP activity in the control group, which showed the highest ALP activity at 72 h (Figure 5A). Loganin decreased the TRAP activity in co-cultured osteoclasts at both 72 and 120 h compared to that in the untreated control group (Figure 5B). Loganin treatment increased ALP activity at 24 h, indicating faster osteogenic activity compared to that in the control group. After 48 h of loganin treatment, ALP activity was reduced, which initiated a change in the bone remodeling communication factors released by osteoblasts. These results indicate that loganin promoted osteoblast differentiation and inhibited osteoclast differentiation in the co-culture system.

### 2.4. Loganin Prevented Ovariectomy (OVX)-Induced Bone Loss in Mice

Estrogen plays an important role in bone remodeling and regulates osteoblast and osteoclast activities [37]. Estrogen increases the functional activity of osteoblasts by inhibiting apoptosis [38]. In addition, estrogen inhibits RANKL-mediated NF-κB activation, which suppresses osteoclast differentiation in monocytes [39]. A decrease or deficiency of estrogen in postmenopausal women triggers an imbalance in bone remodeling, resulting in the development of osteoporosis [1]. The ovariectomized (OVX) mouse is a well-known murine model that mimics estrogen-deficient conditions for the investigation of menopausal osteoporosis in vivo [40]. Therefore, we investigated the effects of loganin on osteoporosis in vivo using an OVX-induced osteoporosis mouse model. Sham-operated or OVX mice were administered loganin (2, 10, and 50 mg/kg/day) for 12 weeks, and the bone mineral density (BMD) of the right femur was examined using a bone densitometer at 0, 6, and 12 weeks. At the end of the 12-week experiment, the trabecular microstructure and bone parameters of loganin-treated mice (50 mg/kg/day) were analyzed using micro-computed tomography (micro-CT). To compare the osteoprotective effects of loganin, 17β-estradiol (E2, 0.03 μg/kg/day) and strontium chloride (SrCl_2_, 10 mg/kg/day) were administered as positive controls. E2 is the most biologically active form of estrogen, and SrCl_2_ is known to exert protective effects in osteoporotic rats [41,42]. As expected, OVX-treated mice presented with significant osteoporotic BMD loss and a hollow trabecular structure (Figure 6A). However, the administration of E2 and SrCl_2_ in positive control OVX mice prevented the decrease in BMD and microstructural loss at 12 weeks (Figure 6A,B). Similarly, the treatment of OVX mice with loganin prevented osteoporotic BMD loss and advanced trabecular microstructures in a dose-dependent manner (Figure 6A,B). In addition, the quantitative analysis of micro-CT imaging showed that the bone volume fraction (BV/TV), trabecular thickness (Tb. Th), trabecular number (Tb. N), and trabecular separation (Tb. Sp) were impaired by OVX induction. However, these parameters were improved by treatment with loganin and the positive controls (Figure 6C). These results indicate that loganin administration protects against OVX-induced bone loss in the mouse femur.

### 2.5. Loganin Increased Serum Osteoprotegerin (OPG) and Decreased Receptor Activator of NF-κB Ligand (RANKL) Levels

OPG is a protein secreted by osteoblast lineage cells that acts as a decoy receptor against RANKL by inhibiting RANK/RANKL-dependent osteoclast differentiation [3]. It has been reported that OPG and RANKL are regulated by estrogen, and the significant postmenopausal reduction in estrogen levels promotes osteoporosis by decreasing and elevating the serum levels of OPG and RANKL, respectively [43,44]. Therefore, the serum OPG/RANKL ratio is closely related to bone mass and osteoporosis development [45]. To determine the effects of loganin on osteoblast and osteoclast communication factors in OVX mice, the serum levels of OPG and RANKL were analyzed using an enzyme-linked immunosorbent assay (ELISA). Postmenopausal osteoporotic OVX mice showed decreased OPG and increased RANKL levels in the serum compared with those of sham-operated mice. However, the loganin treatment ameliorated OVX-induced changes in the serum levels of OPG and RANKL, and consistent results were obtained by treatment with the positive controls E2 and SrCl_2_ (Figure 7A). Loganin significantly increased the OPG/RANKL ratio (Figure 7A), promoting osteogenic activity by reducing osteoclast differentiation and activity, as described for the co-culture system (Figure 7A). In addition, the bone scintigraphy analysis using a single-photon emission computed tomography scan (SPECT scan) was performed. Previously, 99 m-technetium hydroxymethylene diphosphonate (^99m^Tc-HDP) was used for bone scintigraphy analysis because of its high affinity for hydroxyapatite crystals in the bone matrix [46]. Two mice were examined in one panel for comparison purposes. Sham-operated and loganin-treated OVX mice showed an increased ^99m^Tc-HDP uptake compared to that in OVX mice, and similar levels of radioactivity were observed in mice treated with loganin and E2 (Figure 7B). Taken together, these results indicate that loganin inhibits OVX-induced bone loss and promotes osteogenesis by improving bone remodeling markers, such as the OPG/RANKL ratio, in OVX mice.

## 3. Materials and Methods

### 3.1. Cell Culture

The mouse pre-osteoblast cell line MC3T3-E1 (subclone 4, CRL-2593) was obtained from the RIKEN cell bank (Tsukuba, Japan) and maintained in α-modified minimal essential medium (α-MEM) supplemented with 10% Gibco^™^ fetal bovine serum (FBS; Thermo Fisher Scientific, Waltham, MA, USA) and Gibco^™^ Antibiotic–Antimycotic (AA; Thermo Fisher Scientific, Waltham, MA, USA). Bone marrow monocytes were isolated from nine-week-old C57BL/6N mice as previously described [47]. Briefly, the mice were euthanized by cervical dislocation, and the femur and tibia were separated. The bones were disinfected with 100% ethanol, flushed using α-MEM containing 10% FBS without AA, and then filtered through a 40 μm cell strainer to remove debris. The filtered monocytes were then incubated in Petri dishes with α-MEM containing 10% FBS and macrophage/monocyte-colony stimulating factor (M-CSF; Peprotech, Rocky Hill, CT, USA) for 3 days. Isolated monocytes were confirmed using phycoerythrin (PE)-conjugated CD11b antibody (Thermo Fisher Scientific) and a fluorescence-activated cell sorting (FACS) Aria III cell sorter (BD Biosciences, San Jose, CA, USA). Bone marrow monocytes were isolated with permission from the Animal Care and Use Committee of the Ajou University School of Medicine (AMC-133). An in vitro osteoblast–osteoclast co-culture system was established as previously described [48]. Isolated monocytes (4 × 10^4^ cells/well) and MC3T3-E1 cells (2 × 10^4^ cells/well) were seeded in the same plate and incubated in α-MEM containing 10% FBS without AA. All the cultured cells were incubated in a humidified atmosphere at 37 °C and 5% CO_2_.

### 3.2. Water-Soluble Tetrazolium Salt (WST) Assay

Cells (3 × 10^3^ cells/well) were seeded in a 96-well plate overnight and treated with different concentrations of loganin (1, 5, and 10 μM) for 72 h. Cell Counting Kit 8 solution (D-Plus^™^ CCK cell viability assay kit; Dongin Biotech, Seoul, Korea) was added to each well, and the cells were incubated for another 2 h. To determine the cell viability, absorbance was measured at 450 nm using an iMark^™^ Microplate Absorbance Reader (Bio-Rad, Hercules, CA, USA).

### 3.3. Osteoblast/Osteoclast Induction and Examination of ALP/TRAP Activity

To induce osteoblast differentiation, MC3T3-E1 cells were incubated in a complete medium containing 50 μg/mL ascorbic acid and 10 mM β-glycerophosphate for 72 h. Differentiated osteoblasts were solubilized in lysis buffer containing 1 mmol/L Tris–HCl (pH 8.8), 0.5% Triton X-100, and 10 mmol/L Mg^2+^, and ALP activity was determined using p-nitrophenylphosphate (Sigma-Aldrich, St. Louis, MO, USA) as a substrate at an absorbance of 405 nm, according to the manufacturer’s instructions (Bio-Rad, Hercules, CA, USA). To assess the ALP expression in osteoblasts, cells were fixed with 4% paraformaldehyde for 15 min and treated with BCIP/NBT (Sigma-Aldrich, St. Louis, MO, USA) for 30 min at room temperature. The osteoclast differentiation was induced by adding 50 ng/mL of M-CSF (PeproTech) and 50 ng/mL of RANKL (PeproTech) for 5 days. The induction medium was replaced every other day. TRAP activity and staining were determined using an acid-phosphatase kit (Sigma-Aldrich) according to the manufacturer’s instructions.

### 3.4. Quantitative Reverse-Transcription Polymerase Chain Reaction (qRT-PCR)

Total RNA was harvested using QIAzol reagent (QIAGEN, Hilden, Germany) according to the manufacturer’s instructions. Complementary DNA (cDNA) was synthesized using a RevertAid First Strand cDNA Synthesis Kit (Thermo Fisher Scientific). Quantitative reverse-transcription polymerase chain reaction (qRT-PCR) was performed using 100 ng of cDNA and the SYBR Green I qPCR kit (TaKaRa, Shiga, Japan). Fluorescence-based amplification was analyzed using a CFX Connect™ Real-Time System (Bio-Rad). The primers used in this study are listed in Appendix A. The expression levels of each gene were normalized to mouse *Gapdh* (for osteoblasts) and *Hprt* (for osteoclasts), and the relative quantification of the gene expression was calculated using the ΔΔCt method (ΔCt_Treatment_ − ΔCt_Induction_). The fold-change is expressed as 2^−ΔΔCt^.

### 3.5. Ovariectomized Mouse Model

Ovariectomized (*n* = 20) and sham-operated (*n* = 5) ddY mice were obtained from Shizuoka Laboratory Center, Inc. (Hamamatsu, Japan). Mice were maintained in filter-top plastic cages under a controlled atmosphere at 23 ± 2 °C and 55 ± 5% humidity with a 12 h light/dark cycle and provided with food pellets (Feedlab Co., Ltd., Hanam, Korea) and tap water ad libitum. Strontium chloride (SrCl_2_; 10 mg/kg/day, Sigma-Aldrich) and different concentrations of loganin were administered by oral gavage, and 17β-estradiol (E2; 0.03 μg, Sigma-Aldrich) was administered by subcutaneous injection. After 12 weeks of administration, the mice were euthanized and the right femoral bone was harvested for further analysis. All animal procedures were approved by the Animal Care and Use Committee of Ajou University School of Medicine (AMC-133).

### 3.6. Assessment of Bone Mineral Density (BMD) and Micro-Computed Tomography (microCT)

To determine bone mineral density (BMD), mice were anesthetized with tiletamine/zolazepam (Zoletil; Virbac Laboratories, Carros, France) and placed on a specimen plate in the same position. In vivo analysis of the BMD of the longitudinal right femur was performed using a low X-ray energy (80/35 kVp at 500 μA)-emitted PIXI-mus bone densitometer (GE Lunar, Madison, WI, USA). For micro-CT analysis, the transverse right femoral bone was scanned using a Skyscan 1173 micro-CT (Bruker microCT, North Billerica, MA, USA) at the end of the experiment. Three-dimensional axial images were reconstructed, and two-dimensional images were obtained using the NRecon software (Bruker). To quantify the region of interest (ROI) in the femur, including the bone volume (BV/TV) and trabecular number (Tb.N.), thickness (Tb. Th.), and spacing (Tb. Sp.), a region 300 μm below the growth plate was analyzed.

### 3.7. Examination of Serum Levels of Osteoprotegerin (OPG) and Receptor Activator of NF-κB Ligand (RANKL)

At the end of the in vivo experiment, blood samples were collected by cardiac puncture and immediately centrifuged at 1200× *g* for 15 min to obtain the serum. Serum levels of osteoprotegerin (OPG) and a receptor activator of NF-κB ligand (RANKL) were assessed by customized MILLIPLEX^®^ Multiplex Assays using a Luminex^®^ instrument (Merck Millipore, Burlington, MA, USA), following the manufacturer’s instructions.

### 3.8. Single-Photon Emission Computed Tomography Scan (SPECT Scan) Analysis

Mice were anesthetized with isoflurane/N_2_O/O_2_ and intravenously injected with Tc-99m HDP (Mallinckrodt Medical B. V., Petten, The Netherlands). Three hours post-injection, mice were placed prone to the specimen plate, and radioactivity was observed using an Inveon SPECT scanner (Siemens Preclinical Solutions, Malvern, PA, USA).

### 3.9. Statistical Analysis

Data in bar graphs are expressed as the mean ± standard error of the mean (SEM). Statistical analyses were performed using the Statistical Package for the Social Sciences (SPSS 25.0, SPSS Inc., Chicago, IL, USA). Comparisons between multiple groups were determined using one-way analysis of variance (ANOVA) followed by Tukey’s honest post hoc test. Comparisons between two groups were performed using the Student’s *t*-test. Statistical significance was set at *p* < 0.05.

## 4. Conclusions

In the current study, we described the anti-osteoporosis effect of loganin on osteoblast and osteoclast differentiation in vitro and OVX-induced osteoporosis in mice in vivo. Loganin treatment promoted osteoblastogenesis in MC3T3-E1 cells and inhibited osteoclastogenesis in mouse primary monocytes by regulating differentiation-specific markers. The treatment of co-cultured osteoblasts/osteoclasts with loganin revealed that the inhibitory effect of loganin on osteoclast differentiation was induced by increased osteoblast activity, indicating that loganin promotes bone remodeling by increasing osteoblast activity and simultaneously inhibiting osteoclast activity. The in vivo study using OVX mice showed that the treatment with loganin prevented OVX-induced bone loss and improved the serum OPG/RANKL ratio. Our results suggest that loganin may be used as an alternative medicine to protect against osteoporosis.

## Figures and Tables

**Figure 1 ijms-23-14128-f001:**
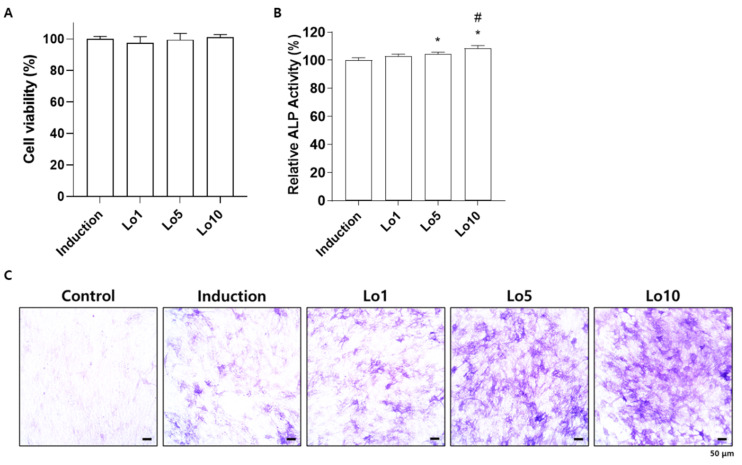
Effects of loganin on the differentiation of mouse preosteoblasts into osteoblasts. MC3T3-E1 cells were incubated in osteoblast induction medium (50 μg/mL of ascorbic acid and 10 mM of β-glycerophosphate) with different concentrations of loganin (1, 5, and 10 μM) for 3 days. (**A**) Cell viability was assessed by Cell Counting Kit 8 (CCK-8) assay. (**B**) Alkaline phosphatase (ALP) activity was determined using p-nitrophenylphosphate as a substrate for ALP. (**C**) ALP-positive cells were visualized under a light microscope. Lo1: loganin 1 μM, Lo5: loganin 5 μM, Lo10: loganin 10 μM. * *p* < 0.05 vs. induction, # *p* < 0.05 vs. Lo1 (one-way ANOVA).

**Figure 2 ijms-23-14128-f002:**
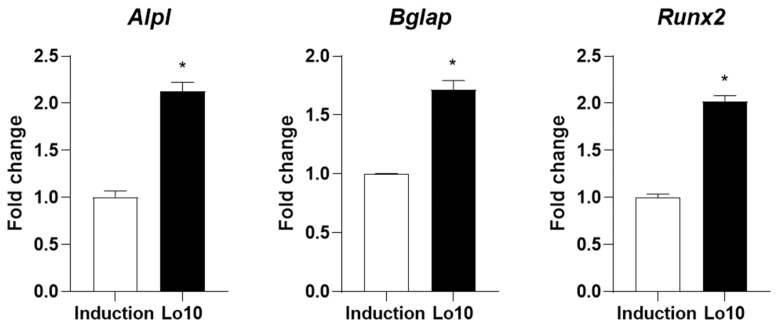
Effects of loganin on osteogenesis-specific markers in mouse preosteoblasts. MC3T3-E1 cells were incubated in osteoblast induction medium (50 μg/mL of ascorbic acid and 10 mM of β-glycerophosphate) with 10 μM loganin for 3 days. The mRNA expression levels of osteogenic markers including *Alpl*, *Bglap,* and *Runx2* were determined by quantitative reverse-transcriptase polymerase chain reaction (qRT-PCR). Lo10: loganin 10 μM. * *p* < 0.05 vs. induction (Student’s *t*-test).

**Figure 3 ijms-23-14128-f003:**
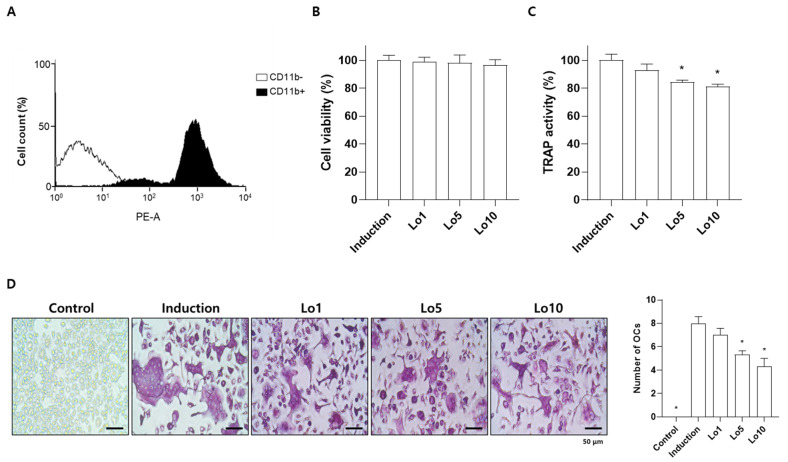
Effects of loganin on the differentiation of mouse bone marrow monocytes into osteoclasts. Bone marrow monocytes were incubated in osteoclast induction medium (50 μg/mL of macrophage/monocyte stimulating factor (M-CSF) and 50 μg/mL of receptor activator of nuclear factor κB ligand (RANKL)) with different concentrations of loganin (1, 5, and 10 μM) for 5 days. (**A**) Monocytes isolated from mouse bone marrow were verified by fluorescence-activated cell sorting (FACS) analysis. (**B**) Cell viability was assessed using a CCK-8 assay. (**C**) TRAP activity was determined using an Acid-Phosphatase Assay Kit. (**D**) Representative tartrate-resistant acid phosphatase (TRAP)-positive cells were visualized by light microscope (left) and the number of osteoclasts were determined (right). PE-A: phycoerythrin-conjugated antibody; Lo1: loganin 1 μM; Lo5: loganin 5 μM; Lo10: loganin 10 μM; OCs: osteoclasts. * *p* < 0.05 vs. induction (one-way ANOVA).

**Figure 4 ijms-23-14128-f004:**
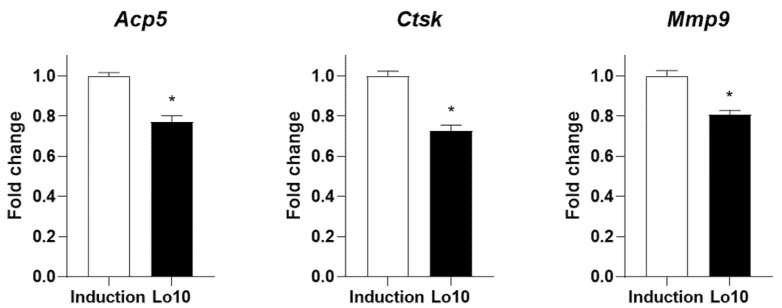
Effects of loganin on osteoclast-specific markers in mouse bone marrow monocytes. Bone marrow monocytes were incubated in osteoclast induction medium (50 μg/mL of M-CSF and 50 μg/mL of RANKL) with 10 μM loganin for 5 days. mRNA expression levels of osteoclast-specific markers including *Acp5*, *Ctsk*, and *Mmp9* were determined by qRT-PCR. Lo10: loganin 10 μM. * *p* < 0.05 vs. induction (Student’s *t*-test).

**Figure 5 ijms-23-14128-f005:**
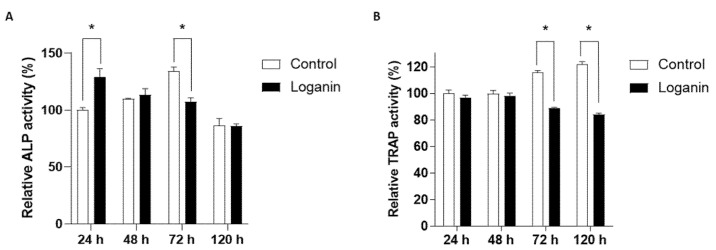
Effects of loganin on osteoblast–osteoclast differentiation in a co-culture system. MC3T3-E1 cells co-cultured with bone marrow monocytes were incubated with loganin (10 μM) for the indicated time periods (24, 48, 72, and 120 h) and the relative (**A**) ALP activity and (**B**) TRAP activity were evaluated. * *p* < 0.05 vs. control (Student’s *t*-test).

**Figure 6 ijms-23-14128-f006:**
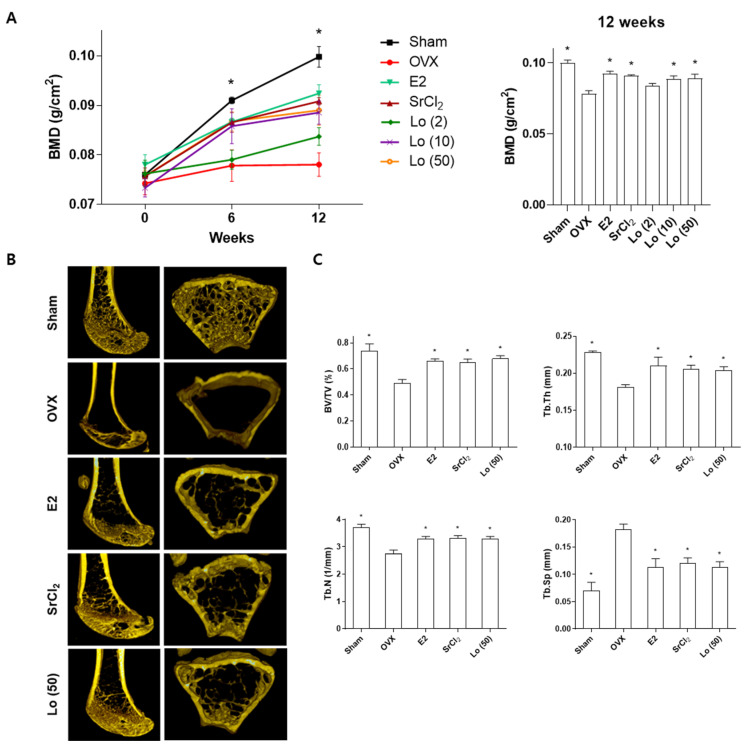
Effects of loganin on ovariectomy (OVX)-induced bone loss in mice. Sham-operated or ovariectomized (OVX) Deutschland, Denken and Yoken (ddY) mice were administered 17β-estradiol (E2, 0.03 μg/kg/day), strontium chloride (SrCl_2_, 10 mg/kg/day), or loganin (50 mg/kg/day) for 12 weeks. (**A**) The bone mineral density (BMD) of the right femur was assessed at the indicated time periods (0, 6, and 12 weeks) by a PIXI-mus bone densitometer. (**B**) The microstructure and (**C**) bone parameters of the femur were examined by micro-CT analysis. * *p* < 0.05 vs. OVX (one-way ANOVA).

**Figure 7 ijms-23-14128-f007:**
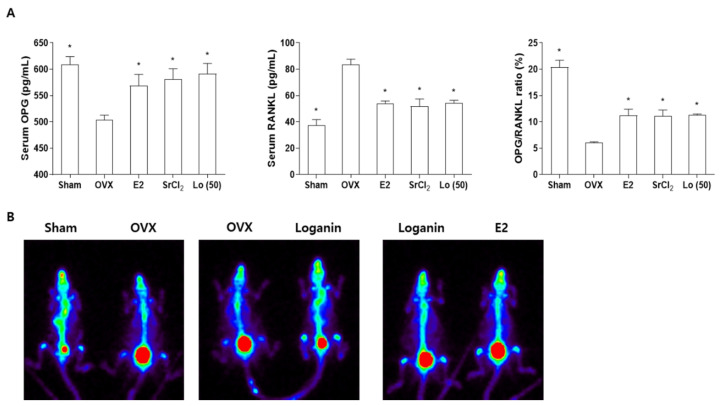
Effects of loganin on ovariectomy (OVX)-induced parameters in the serum and bone of mice. Sham-operated or ovariectomized (OVX) ddY mice were administered 17β-estradiol (E2, 0.03 μg), strontium chloride (SrCl_2_, 10 mg/kg/day), or loganin (50 mg/kg/day) for 12 weeks. (**A**) Serum levels of osteoprotegerin (OPG) and receptor activator of nuclear factor-κB ligand (RANKL) were examined using a MILLIPLEX^®^ multiplex assay. (**B**) Bone scintigraphy images using technetium Tc-99m HDP (^99m^Tc-HDP) were obtained by Inveon SPECT scan. * *p* < 0.05 vs. OVX (one-way ANOVA).

## Data Availability

The data presented in this study are available on request from the corresponding author.

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
