# Peer review of "Effects of Loganin on Bone Formation and Resorption In Vitro and In Vivo"

_ijms, 2022, doi:10.3390/ijms232214128_

Round 1

Reviewer 1 Report

The authors present an extremely intriguing manuscript on Effects of Loganin on Bone Formation and Resorption in vivo and in vivo. The manuscript is well-written, and the experiment to demonstrate the effect of Loganin on bone cells and bone metabolism is well-designed. The numbers are generally transparent (exept for figure 7, comment here below). Thus, I am in favour of the manuscript and believe that a few minor modifications would make it publishable and of interest to bone biologists. 

Line 41. The term osteocyte is typically reserved for a subset of obsteoblasts embedded in the bone matrix and not for "bone cells." 

Line 53. Long-term antiresorptive drug use is associated with atypical femoral fractures and osteonecrosis of the jaw, which I believe are the greatest obstacles. In addition, we have a new agent that targets sclerostin and PTH. 

Were concentrations of Loganin above 10 M evaluated, as it appears that the dose-effect curve was not at its plateau? 

Fiigure 3 D. I would have liked the number of osteoclasts to be determined so that the effect of Loganin on osteoclastogenesis could be confirmed. 

The figure 7 legend does not correspond to the figure itself. B) represents the SPECT scan, and there is no C in the illustration.

Author Response

Please see atteched file.

Reviewer 2 Report

Manuscript entitled "Effects of Loganin on Bone Formation and Resorption in vitro  and in vivo" by  Chang-Gun Lee  et al. describes the effects of loganin, an iridoid glycoside, on bone formation to treat osteoporosis by demonstrating both in vitro and in vivo" Please find the comments below:

1) The authors must pay attention to the title: it is spelled in vivo instead of in vitro. Please correct it and double check to proofread the manuscript once again to make any typos.

2) Unlike the date shown on Fig. 7b (in vivo), why didnt the authors considered E2 to test in in vitro? Please explain and make a coherent explanation for in vitro and in vivo study comparison.

3) Fig 6a is not clearly labelled, please revise the graph with different symbols of lines with different colors. the present graph doesnt show clear differences.

4) For fig 7b, please combine sham, OVX, loganin and E2 in a single image if they are acquired on the same day or different images with different days to distinguish the differences with same signal intensity. please show the scale next to image. The current image doesn't show the intensity scale.

Please address as a minor revision.

Author Response

Manuscript entitled "Effects of Loganin on Bone Formation and Resorption in vitro and in vivo" by  Chang-Gun Lee  et al. describes the effects of loganin, an iridoid glycoside, on bone formation to treat osteoporosis by demonstrating both in vitro and in vivo" Please find the comments below:

1) The authors must pay attention to the title: it is spelled in vivo instead of in vitro. Please correct it and double check to proofread the manuscript once again to make any typos.

We corrected spelling of the word to “in vitro” in the title and checked mistyping throughout the manuscript again according to the reviewer’s comments.

2) Unlike the date shown on Fig. 7b (in vivo), why didnt the authors considered E2 to test in in vitro? Please explain and make a coherent explanation for in vitro and in vivo study comparison.

Thank you very much for your kind advice. We agreed with reviewer’s comment of a coherent explanation for in vitro and in vivo study comparison. In addition, several studies suggested that E2 on osteoporosis experiment in vitro have been used as a positive control. We regret to focus on osteoblast and osteoclast differentiation in vitro without positive control and treatments of SrCl2 and E2 might be more appropriate in vivo model as positive controls. We will consider the coherent investigation of E2 for in vitro and in vivo study comparison for further experiment.

3) Fig 6a is not clearly labelled, please revise the graph with different symbols of lines with different colors. the present graph doesnt show clear differences.

As per the reviewer’s comments, we revised the graph with different symbols of lines and colors, including Y-axis to make the clear differences between groups.

4) For fig 7b, please combine sham, OVX, loganin and E2 in a single image if they are acquired on the same day or different images with different days to distinguish the differences with same signal intensity. please show the scale next to image. The current image doesn't show the intensity scale.

Mice were measured the skeletal uptake of technetium (Tc-99m) HDP on planar HDP bone scans in Department of Molecular Imaging, Korea Institute of Radiological and Medical Sciences. There was restricted time (a day) and use of SPECT scanner to minimize exposure time of radioactivity during the examination period for the safety. For this limitation, instructor recommended 3 paired samples only in a day. Scanned radioactive images represented skeletal blood flow and osteogenic activity, and the levels of radioactivity suddenly disappeared. For accurate injection and scanning time, paired mice were scanned only once.

Please address as a minor revision.